# The Digital Divide and Active Aging in China

**DOI:** 10.3390/ijerph182312675

**Published:** 2021-12-01

**Authors:** Lingchen Liu, Fan Wu, Huiying Tong, Cuihong Hao, Tingting Xie

**Affiliations:** 1School of Statistics, Shanxi University of Finance and Economics, Taiyuan 030006, China; liulingchen@fudan.edu.cn (L.L.); wufan1224@stu.sxufe.edu.cn (F.W.); tonghuiying26@stu.sxufe.edu.cn (H.T.); 20181083@sxufe.edu.cn (C.H.); 2Center for Population and Development Policy Studies, Fudan University, Shanghai 200433, China; 3Department of Global Health and Population, Harvard TH Chan School of Public Health, Boston, MA 02115, USA; 4School of Economics, Peking University, 5 Summer Palace Road Street, Beijing 100871, China

**Keywords:** digital divide, active aging, entropy weight method

## Abstract

With the rapid development of society, especially the advent of intelligent technology of life services, the ability of the elderly to adapt to modern digital life is getting weaker and weaker, the dilemma of the “digital divide” for the elderly has aroused heated public debate. In the era of aging and information superposition, in-depth study of the multi-dimensional impact of the digital divide on the elderly has become an effective content of China’s active aging strategy. Based on the micro-data of the Chinese General Social Survey in 2017, this paper uses the entropy right method to construct the digital divide index system for the elderly over 60 years of age from the perspective of essential equipment, Internet utilization degree, and Basic Internet usage skills. At the same time, this paper constructs China’s comprehensive index of active aging from three aspects: health, social participation, and guarantee of the elderly, to study the impact of the digital divide on China’s active aging. The following conclusions have been drawn: the digital divide among the elderly significantly inhibits China’s active aging. The digital divide reduces the level of physical and mental health and social participation of older persons and inhibits the level of guarantee of older persons, thus impeding their active aging. In addition, it also reduces the overall life satisfaction of the elderly. The use of the Internet, skills, and other digital technology abilities of the elderly have effectively promoted active aging. The more Internet access devices older people have, the higher their level of social participation. The higher the Internet frequency of the elderly, the healthier the body and mind. Furthermore, the greater the level of physical and mental health and social participation of older groups who use online payments. The digital divide among the elderly inhibits the process of China’s active aging, and the unique course and stage characteristics of the development of the aging of the Chinese population require us to pay full attention to the relationship between the digital divide and active aging and how to construct a “digital-friendly” aging system is an essential issue for China’s social development to consider.

## 1. Introduction

With the advent of the digital age, the world economy has radiated new vitality. The digital economy has increasingly become a new driving force for economic and social development. In recent years, China’s digital economy has developed rapidly. The increasing integration of a new generation of digital technology represented by the Internet, big data, cloud computing, artificial intelligence, blockchain, and the Internet of things with the real economy has injected a new source of China’s economic growth. According to the white paper on the development of China’s digital economy (2021), in 2020, the added value of China’s digital economy will reach 39.2 trillion yuan, accounting for 38.6% of GDP. The achievement of digital development has also brought a real sense of gain to ordinary people. Especially after the outbreak of COVID-19 in early 2020, people’s production, life, and working methods have undergone profound changes. Now, most people have become more dependent on digital technology than ever before. Intelligent manufacturing, online education, cloud medical treatment, cloud office, cloud display, online entertainment, and online consumption have become typical scenes of Digital China. At the same time, the trend of population aging in China is more prominent. According to data from the Seventh National Census, in 2020, China’s total population aged 65 and over was about 191 million, accounting for 13.5% of the total population [1]. China began to enter the stage of moderate aging gradually. According to the prediction of the United Nations, by the middle of the 21st century, the scale of China’s elderly population will continue to expand, and the aging process will continue to accelerate and deepen. It is estimated that by 2035, China’s population aged 65 and over will exceed 300 million and finally reach the peak of the 21st century around 2058, with about 385 million people accounting for about 28.0% of the total population [2]. Under the dual background of digitization and aging, the social governance problem of China’s elderly digital divide will become more prominent. Due to the constraints of technology, system, culture, and the elderly themselves, there are significant differences in the ownership and application of information technology between the elderly group and other groups [3]. For example, many older people do not have intelligent devices, cannot access the Internet, download APPs, obtain the required information, and are at a loss in the face of complex operating procedures. These will undoubtedly bring great inconvenience to the daily life of the elderly so that they cannot fully enjoy the convenience brought by digital and intelligent services, are excluded from the digital society, and become “digital survivors” or “digital refugees”. Especially in the public health crisis, which was caused by the outbreak of COVID-19 in early 2020, because of Internet barriers, it is difficult for the elderly to make full use of mobile phones and other smart devices to get essential information resources related to the epidemic in time. It makes them frequently in a dilemma, and the public health crisis has further transformed into the information crisis of the elderly. The importance and urgency of digital divide governance have become more and more prominent.

Under the background of China’s accelerated aging and the continuous expansion of the scale of the elderly population, how to deal with the challenges of aging and realize active and healthy aging is not only the focus of discussion in academic circles and government departments but also the focus of concern from all walks of life. So what is active aging? The concept of active aging officially put forward by the World Health Organization at the second world assembly on aging in 2002 is an upgrade of the concept of healthy aging, emphasizing the health, social participation, and security of the elderly. Health refers to the condition of maintaining positive health in physical, mental, and spiritual aspects, including physical health, mental health, and good social adaptation; “Participation” means that the elderly participate in socio-economic, cultural, and spiritual activities according to their abilities, needs, and preferences; “Security” refers to supporting families and communities to take care of the elderly through various ways when they are unable to take care of themselves [4,5].

Accelerating the digital integration of the elderly and narrowing the digital divide of the elderly are the new needs and actual contents of promoting active aging in the digital society. How to narrow the digital divide of the elderly in the digital age and promote active aging policies has attracted the country’s great attention. On 24 October 2020, the general office of the State Council issued the implementation plan on effectively solving the difficulties of the elderly in using intelligent technology. The plan focuses on seven types of high-frequency matters and service scenarios involved in the daily life of the elderly, such as travel, medical treatment, consumption, entertainment, and affairs, and effectively solves the problem of the digital divide faced by the elderly by establishing a long-term mechanism. To further deepen the understanding of the impact of the elderly digital divide on active aging and enhance the effectiveness and pertinence of the governance of the elderly digital divide, this paper first constructs a comprehensive index of the digital divide and a comprehensive index of active aging, and then quantitatively analyzes the impact of the elderly digital divide on active aging. It has significant theoretical value and practical significance for the research in China, which is still a weak field and has reference value for the formulation of aging digital policies suitable for China’s active aging.

## 2. Literature Review

The digital divide originated from American media reports and government announcements in the 1990s. Its concept is traditionally defined as the gap between technology access owners and technology access lack [6]. The digital divide theory emphasizes that the differences in the accessibility and use of the Internet among different groups in society lead to the digital divide [7], which is divided into access ditch and use ditch according to different causes. The condition gap of different people accessing the Internet at the material level is called the first digital divide, namely access ditch [8]. The access ditch mainly reflects the inequality of material access based on economic and social development, mainly affected by national economic strength, government decision-making, network infrastructure construction, and information technology standards and norms [9,10]. However, having the same hardware access conditions does not mean that people will use the Internet in the same way and degree. This structural difference in digital skills and use is the second digital divide the use ditch. The research on the Internet use ditch mainly focuses on the gap of online time between different groups, the purpose of using the network, and the content of network online activities [11,12]. According to different levels of digital skills, digital skills can be divided into tool skills, information skills, and strategy skills [13]. These three skills are not evenly distributed in the whole social population. Some scholars have studied that the digital divide between the elderly and the young goes far beyond the simple access problem. Due to physical and cognitive limitations, low computer and information literacy, barriers and fears to technology adoption and application, the elderly lag behind the young in the use of information technology, which is a common phenomenon all over the world [14,15,16,17], and forms an old-age digital divide due to age [18,19]. In 2019, the elderly population aged 65 and above accounted for 16.2% in the United States. At the same time, the 15th edition of the Internet Usage Survey Report released by the National Telecommunications and Information Administration (NTIA) in 2020 shows that as of November 2019, nearly 80% of Americans used the Internet, but the digital divide that existed among the elderly was still large, and the proportion of people over 65 who used the Internet was less than 70%. In 2019, the proportion of the elderly population aged 65 and over in the UK reached 18.51%. According to a data report released by the Office of National Statistics (ONS), more than half of the respondents aged 65 and over said they had shopped online. In 2019, Russia had 16.2% of the population aged 65 and above and 36% of the population aged 65 and above used the Internet, while South Korea had 14.9% of the population aged 65 and above, 38.9% of the population aged 70 and above used the Internet, and 37.8% of the population used smartphones. According to the National Bureau of Statistics, 13.5% of China’s population was over 65 in 2020. As of March 2020, China’s elderly Internet users over 60 years old accounted for 6.7% of the total number of Internet users, and the penetration rate of elderly Internet users was 23.7%, less than 1/3 of the young Internet users (73.0%) (estimated according to the total population and composition of China at the end of 2019). On the user side, due to the weakness of digital skills, the proportion of the elderly in search engines, APP installation, and WeChat use was significantly lower than that of young people in China. Among them, the proportion of the elderly using search engines was 4.4%, less than 1/6 of the non-elderly Internet users (27.4%). The number of mobile APPs per capita for the elderly was 37, which was only 44.0% of the number of young Internet users aged 20–29 (84 per capita); The proportion of the elderly using WeChat was 26.2%, less than 1/3 of the non-elderly users (88.9%) [20]. Although the aging level of China is relatively low, due to a large base and a large elderly population, the digital divide of the elderly is prominent. So measures taken by other countries and regions to deal with the digital divide of the elderly have certain reference value and significance for China. To sum up, this paper defines the digital divide of the elderly as the gap between the elderly group and the middle-aged and young group in using and obtaining information due to their disadvantages in equipment acquisition, the level of media use, and basic user skills. The existing literature on the digital divide of the elderly is primarily qualitative and lacks quantitative analysis. As a first innovation, this paper attempts to construct the elderly digital divide quantitatively from the gap of digital ability between the elderly and young people.

In addition, previous literature thinks whether to use the Internet is an important indicator reflecting the digital divide [21,22]. Furthermore, in relevant research, scholars mostly used the difference in Internet use to reflect the digital divide. As for measuring the digital divide among the elderly, the index of whether to use the Internet was mostly used in the literature [23,24]. Some scholars measured the Internet use of the elderly from three dimensions: whether to use the Internet, the amount of online content, and frequency of use [5], but it is still tricky to reflect the digital technology use of the elderly comprehensively. This paper attempts to construct a comprehensive index of the digital divide, thereby comprehensively analyzing the impact of the digital divide on active aging.

Most of the existing literature only examines the impact of Internet use of the elderly on their physical and mental health and social participation. In terms of physical and mental health, some scholars believe that the elderly can master more health knowledge through the Internet and then improve their physical health level [25,26]. Internet use can also reduce the loneliness of the elderly [27,28,29,30], accordingly improving their mental health [31,32,33,34]. Furthermore, some studies have shown that Internet use can also promote the mental health development of the elderly by increasing their learning frequency [35]. In terms of social participation, some scholars found that using the Internet can promote the overall social participation of the elderly [36] and improve the overall life satisfaction of the elderly [5]. Most of the previous literature studied from a single aspect of active aging, but active aging includes physical and mental health, social participation, and security. Therefore, there is a lack of comprehensive research on the impact of active aging. This paper attempts to construct a comprehensive index of active aging from the three dimensions of physical and mental health, social participation, and security, thereby studying the comprehensive impact of the digital divide on active aging.

To sum up, this paper first constructs the comprehensive indicators of the digital divide and active aging and then quantitatively analyzes the impact of the elderly digital divide on active aging. In addition, starting from the three secondary indicators of essential equipment, Internet utilization degree, and basic Internet use skills, and from the perspective of more specific tertiary indicators, such as the number of Internet devices, Internet frequency, online reading time, whether they can make electronic payments, open web pages and search for information, this paper better investigates the role of different types of activities and intensity of the digital divide, and finally, get some important policy enlightenment according to the empirical results.

## 3. Materials and Methods

### 3.1. Data

This study used cross-sectional data from the 2017 China General Social Survey Data (CGSS). Considering that the supplementary method will significantly reduce the accuracy of the data, this paper used the culling method to delete the missing samples of critical variables of the elderly aged less than 60 years and over, and the final sample size reached 298 people, using a total of 26 specific indicators in the study.

### 3.2. Methods

#### 3.2.1. Dependent Variable

Active aging includes three main categories of health, participation, and guarantee; this paper not only studied the impact of the digital divide on health, social participation, and guarantee at the individual level of the elderly, but also paid more attention to the impact of the digital divide on active aging. Life satisfaction can reflect the development of active aging in the elderly population from the side, so this paper verified the impact of the digital divide on active aging through life satisfaction. Based on the CGSS survey data, this paper’s physical and mental health dimension was selected from both physical and mental aspects. The physical health index was “physical health,” the mental health index was “whether life is happy”, “feeling lack of companionship”, “isolated frequency”, and “whether you will feel lonely” are four indicators; each indicator is valued at 0 or 1. Learn from Jin Yongai [5], and this paper added up to five numerical indicators, score 0–5 points, the higher the score indicates the better physical and mental health. The dimension of social participation was the same, mainly from the types of social activities involved by the elderly groups, including watching movies, shopping, participating in activities and gatherings with relatives and friends. Of the frequency of 5 indicators, the value range was 5–25, 5 indicates a weak sense of social participation, 25 indicates a strong sense of social participation. The guarantee selection indicator was whether to participate in endowment insurance or medical insurance. If the elderly participate in at least one of them, it is 1; otherwise, 0. In addition, life satisfaction as a comprehensive indicator can reflect the overall life satisfaction of the elderly groups, the value range of 1–5, 1 indicates very dissatisfaction with life, 5 indicates that life is very satisfying.

After constructing a comprehensive evaluation index system for active aging, the next step was to empower each index. The size of the indicator’s weight directly determined the importance of the indicator and affected the final result. Therefore, how to choose a scientific and effective method of empowerment is very important. At present, the commonly used methods of empowerment mainly include personal empowerment and objective empowerment.

The subjective empowerment method is based on expert opinion, which mainly depends on the expert’s experience and knowledge reserved for many years. However, subjective empowerment law relies too much on people’s subjective judgment, and the quality of experts is uneven, easy to affect the evaluation results. The subjective empowerment law mainly includes hierarchical analysis and expert opinion.

The objective empowerment method is based on the intrinsic relationship of objective values to calculate the index weight of the empowerment method, not affected by subjective judgment, and the result is more objective than others. Commonly used objective empowerment methods include factor analysis, principal component analysis, neural network, entropy.

Compared with other empowerment methods, the entropy power method can make full use of systematic information, consider the ambiguity and uncertainty of evaluation criteria and indicators, and effectively reduce the subjectivity in evaluation. Therefore, this paper used the entropy right method as the empowerment method of the positive aging comprehensive evaluation, calculated the final score, and made the evaluation result more accurate.

Active aging comprises three indicators: physical and mental health, social participation, and security, synthesized by the entropy weight method, and the higher the value, the higher the degree of active aging of the elderly population. The sample of 298 elderly persons was selected, and three evaluation indicators were designed, namely, “health”, “social participation”, and “guarantee”. $x_{ij}$ represented the j-th evaluation indicator value of the i-th sample (i = 1,2,3…298, j = 1,2,3). The application of the entropy method is as follows:

(1) The original data is processed without a quantity outline, eliminating the physical scale’s influence and calculating the i-th sample’s contribution under the j-th index (Equation (1)).
(1)pij=xij∑i=1289xij

(2) Calculate the entropy value. Calculate the entropy value of the j-th indicator (Equation (2)).
(2)$$e_{j}=-\frac{1}{289}\sum_{i=1}^{289}p_{ij}\ln\left(p_{ij}\right),\;0\leq e_{j}\leq 1$$

(3) Calculate the difference coefficient (Equation (3)).
(3)$$g_{j}=1-e_{j}$$

(4) Determine the weight of the evaluation index and calculate the total score of each sample (Equation (4)).
(4)Wj=gj∑j=13gj, j=1,2,3

By calculation, the weight of “health” was 0.246, the weight of “social participation” was 0.668, and the weight of “guarantee” was 0.086.

#### 3.2.2. The Core Independent Variable

This study measured the ability to use digital technology for the elderly population from multiple dimensions, constructed its digital ability index system, and then calculated the digital divide of the elderly. This study took the definition of the digital divide as the starting point to measure the personal digital ability index of the elderly group from the three dimensions of “basic equipment”, “Internet usage level”, and “Internet basic ability”, and under these three secondary indicators, nine three-level indicators were set up. The three-level index corresponding to “basic equipment” was the number of Internet access devices. The three-level indicators corresponding to “Internet usage level” were whether to surf the Internet, online reading time, and online frequency. The value range of the two-level indicators was 5–15. The larger the value, the higher the frequency of using the Internet. The three-level indicators corresponding to “essential Internet ability” were whether they can pay online, open web pages, download, find information online, and contact relatives and friends online. The value range was 0–5. The larger the value, the more basic Internet ability they master. The three-level indicators were standardized and added to obtain the second-level indicators, and then the first-level indicators of digital capability were synthesized by the entropy weight method.

According to the calculation, the weight of “basic equipment” was 0.403, “Internet utilization” was 0.245, “basic Internet useability” was 0.352, and the digital divide of the elderly was obtained by comparing the digital ability of the middle-aged and young people with the digital ability of the elderly.

#### 3.2.3. Control Variables

The essential demographic and socio-economic characteristics of the elderly were controlled in the regression model, including age, sex, education level, marital status, account, political outlook, number of real estates, region, living alone, cognitive ability. Among them, gender, account, the political outlook, whether to live alone were two variables. Due to the limited sample data, the cognitive ability index was selected from the language ability, and the value range was 2–10. It can be seen from Table 1 that the proportion of men and women in the selected survey samples was relatively balanced, with more than 3/4 in the eastern region and 78% in the married group. Most of the elderly had a political outlook of the masses, and less than 1/5 of the elderly lived alone, with moderate years of education, with an average of 10.88 years. The basic information of variables can be seen in Table 1.

#### 3.2.4. Policy Analysis

In order to facilitate the explanation, the least square method (OLS) was used to make a linear regression analysis of the indicators of health, social participation, and active aging for the digital divide and digital ability, and the ordered-probity method was used for regression life satisfaction and security as classified variables to preliminarily explore the relationship between the digital divide and the indicators of active aging of the elderly. In addition, this paper further analyzed the impact of the secondary and tertiary indicators of the digital ability of the elderly on the indicators of active aging.

## 4. Results Analysis

Table 2 shows the relationship between the digital divide of the elderly and active aging, the indicators of active aging, and life satisfaction under the control of other variables. The regression results showed that the larger the digital divide, the less protection the elderly groups got, and the weaker their sense of physical and mental health and social participation, so that the realization of active aging was restrained, and the life satisfaction of the elderly groups weakened, which further confirms the impact of the digital divide on active aging. Specifically, every unit of the digital divide increased, the social security decreased by 0.467 units, the physical and mental health status and social participation of the elderly decreased by 0.498 points and 0.316 points, respectively, the active aging decreased by 0.335 units, and the satisfaction of the elderly in life decreased by 0.164 units. The above is because different age groups have different degrees of acceptance of digital technology. Young and middle-aged people accept digital technology more quickly. They actively learn digital skills so that they can maximize the convenience given by its dividend. Due to age, living environment, education, and other reasons, it is difficult for the elderly to integrate into the rapidly developing technological society quickly. As a result, with the development of digital technology, the gap between the elderly and young people in the ability to use digital technology has only increased, resulting in a digital gap that cannot be ignored. Such scenes as inconvenient travel without health code, inability to get a doctor’s number online, and complex online payment continue to appear, which makes many elderly groups experience difficulty in their daily lives and inhibits the process of active aging of the elderly.

Other control variables were also significantly correlated with the indicators of active aging of the elderly. Compared with the unmarried and the elderly whose partners died; the married elderly group felt more difficulties in life. Perhaps due to the decline of physical function, unmarried people may gradually feel the inconvenience of being alone after entering their 60s, and widowed older people found difficulty adapting to the transition from having help to having to take care of themselves. In addition to the decline of physical health, the death of their partner affected their mental health to a great extent. Divorce had a significant positive impact on the social participation of the elderly, which may be due to the lack of spouse constraints; the elderly will be more inclined to participate in various social activities to enrich their lives. The degree of active aging of unmarried and widowed elderly groups was significantly lower than that of married elderly groups. In addition, the number of real estates reflected the socio-economic status of the elderly, and the health status of the elderly with more real estate was better than others. Compared with the masses, the elderly groups with a political outlook of Communist Party members had better health and social participation. Compared with rural hukou, the elderly groups with non-rural Hukou had a stronger sense of social participation, which may be due to the higher types of leisure and entertainment life in urban areas than in most rural areas, and the differences in consumption views between urban and rural elderly groups. From the perspective of geographical location, the life satisfaction of the elderly group in the central and eastern regions was significantly higher than that in the western region, and the health status was significantly better than that in the western region. The possible reason is that the better the regional economic development, the better the life status of the elderly group. Cognitive ability is essential in the environment of old age, and from the results, the higher the cognitive ability score, the fewer difficulties older people experience in life that cannot be solved.

Table 3 shows the impact of secondary indicators of digital ability—base equipment, Internet usage level, and essential ability on active aging, indicators of active aging, and life satisfaction of the elderly, respectively, under the control of other variables. The greater the number of base equipment owned by the elderly, the stronger the sense of social participation and guarantee level, the higher the degree of active aging, and the stronger the life satisfaction. The higher the degree of Internet use, the better the health status and the stronger the life satisfaction. It is noteworthy that the level of essential Internet useability of the elderly had a significant impact on active aging and its indicators. The higher the level of basic Internet use skills, the stronger the elderly group’s participation in social security, health level, and sense of social participation, the better the overall situation of active aging, and the more vital life satisfaction. The regression results showed that social participation increased by 0.291 points for each additional essential equipment, guarantee, and active aging by 0.684 and 0.24 units, respectively, and life satisfaction increased by 0.21 units. For every increase in Internet utilization, health increased by 0.225 points, active aging increased by 0.141 units, and life satisfaction increased by 0.064 units. The essential ability had a significant impact on various indicators of active aging. An increase of 1 unit for active aging increased security by an average of 0.391 units, health and social participation increased by 0.405 and 0.237 points, respectively, active aging increased by 0.767 units, and life satisfaction increased by 0.085 units.

Table 4 shows the impact of the three-level indicators of digital ability on the indicators of active aging and the life satisfaction of the elderly under the control of other variables. These indicators were the number of Internet devices, Internet frequency, online reading time, whether to make an electronic payment, open web pages, and search information. All the above three indicators had a significant negative impact on the life satisfaction of the elderly group. The frequency of surfing the Internet and the time of reading on the Internet had a significant positive impact on the health status, social participation, and social security of the elderly. The more Internet access devices the elderly had, the stronger the sense of social participation and the higher the degree of active aging. It is noteworthy that the health status, social participation, and active aging of the elderly who could use electronic payment were significantly better than those who could not. Similarly, the elderly who could open a web page performed better in various indicators of active aging. Compared with the elderly who could not find information on the Internet, the elderly who could find information had a higher degree of health, social participation, security, and active aging.

## 5. Conclusions

### 5.1. Conclusions

There was a significant negative correlation between the digital divide among older persons and China’s active aging. The greater the digital divide between the elderly and young and middle-aged people, the lower the level of active aging of the elderly group. For every 1 unit increase in the digital divide, the level of active aging of the elderly decreased by 0.335 units. By breaking down the digital divide, it was found that there was a positive correlation between indicators of the digital technological capacity of older persons and active aging. The higher their utilization of the Internet, the more operational skills they mastered, and the higher their level of active aging. At the same time, the digital divide also increased the frustration of the elderly. The larger the digital divide, the elderly often felt many difficulties.

Family, location, and other factors also significantly influenced the level of active aging of the elderly group. Widowhood had a significant inhibitory effect on the physical and mental health of the elderly, which led to a reduction in the active aging level of the elderly to a great extent. In addition, the sense of social participation of the divorced elderly group was significantly more potent than that of other older people. The elderly who were political members of the Communist Party were in good health and social participation, and the level of active aging was higher than that of the elderly with a political outlook of the masses. The active aging level of the elderly with non-agricultural household registration was significantly higher than that of the elderly with agricultural household registration, which showed that there was also a large gap between urban and rural areas.

The more Internet access devices the elderly have, the higher their level of social participation and the higher their degree of active aging. The higher the Internet frequency of the elderly, the more information they receive, making them happy, improving their physical health, and promoting active aging. If the elderly can use online payment, it will significantly enhance their physical and mental health level and sense of social participation. The security level of the elderly who can open web pages and find information on the Internet is much higher than that of other elderly groups, which will promote the active aging level of these elderly groups. The study finds that the higher the digital technology mastery ability of the elderly group, the stronger their life satisfaction. On the other hand, it reflects the improvement in the active aging level of the elderly group.

### 5.2. Recommendations

The digital divide between the elderly and the middle-aged and young people makes it difficult for the elderly to enjoy the dividends of the digital age fully, share the fruits of the development of the digital economy, and maximize society’s overall welfare. It brings adverse effects on the physical and mental health, social participation, and security of the elderly, thus not conducive to active aging. Therefore, the family, government, and society should work together to actively implement digital feedback, strengthen digital security, launch aging products, encourage the elderly to participate in digital life actively, and expand the consumption of digital products, to realize the social universality of digital economic well-being.

(1) Implement family digital feedback and improve the digital participation of the elderly

The Network Center for the elderly in the United States advocates the use of intergenerational interaction to guide the elderly to master basic computer skills by high school students and college students. Under the guidance of this method, the elderly signed up to participate in the project of young people teaching digital skills to the elderly, which better improved their digital literacy and skills [37]. This measure provides a useful reference for our country. Therefore, in order to narrow the digital gap among the elderly group in China, we can implement family digital feedback, encourage the young members of the family to teach the elderly the knowledge related to digital technology, stimulate the learning interest of the elderly group, drive the elderly to participate in the digital life and enjoy the welfare of the digital society. It can be seen from the conclusions that the ability to use electronic payments, open web pages, and search for information on the Internet can all promote the improvement in the level of active aging of the elderly. Therefore, family members can start from the following aspects: first, popularize the basic Internet operation skills, such as how to download some commonly used APPs, how to use mobile phones for online payment, how to contact and communicate with family and friends through mobile phone applications, and how to find the required information on the Internet. Second, popularize network security knowledge and improve their network security awareness, such as how to distinguish the authenticity of network information and how to protect personal information security on the network, guide them to protect personal information when surfing the Internet and avoid information disclosure and personal privacy infringement. The third is to create an excellent digital technology learning environment, patiently and carefully impart relevant knowledge to the elders, give them psychological and action support, be tireless when the elderly doubt, understand and tolerate when they make mistakes, and become an active guide, enthusiastic supporter and patient caregiver for the elderly on the Internet. This process not only strengthens the interaction between family members, enhances the understanding and emotion of the younger generation towards their elders, narrows the intergenerational gap, but also helps the elderly learn the relevant knowledge of the Internet and the essential operation of digital technology, narrows the digital gap between the elderly and promotes the realization of active aging.

(2) The government plays a central role in narrowing the digital divide among the elderly

The US government provided fund support in public infrastructure to narrow the gap in Internet coverage, and social groups carried out technical training for the elderly to promote their digital literacy [37]. The German government launched an Internet strategy for the elderly, aiming to increase the Internet penetration rate of the elderly from less than half at present to 80–90% in the next 10 years. The Singapore government also set up a Digital Transformation Office in 2020, recruiting 1000 “digital ambassadors” to reach out to communities to help about 100,000 seniors acquire digital skills [38]. In addition, in order to solve the problem of “information silos” among the rural elderly, the US federal and state governments identified rural broadband access as a priority funding project, providing training and support for the rural elderly and helping them gain more and better access to Internet technologies and services [37]. Therefore, in order to accelerate the digital integration of the elderly and narrow the digital gap between the elderly, these measures taken by foreign governments are also worth learning from. First, public institutions should strengthen the publicity and guidance on using digital technology for the elderly. Television, newspapers, and other traditional media are the main channels for the elderly to obtain information, and they have a high degree of dependence and trust in them. Therefore, the government can make full use of this advantage to publicize the convenience of using digital technology and the richness of access to information in traditional media, support and encourage the elderly to use the Internet for reading and socializing, and then stimulate the elderly’s interest in learning in new fields. At the same time, it also needs to publicize relevant network security knowledge and improve their awareness of online security. Second, strengthen network security supervision, build a network information security defense system, severely crack down on network fraud and illegal acts infringing on others’ information security, introduce corresponding systems to ensure that the elderly group can safely integrate into Internet life, and then promote the realization of their active aging. Finally, narrow the gap between urban and rural areas, promote the development of urban–rural integration, and then narrow the digital divide between urban and rural elderly, and finally realize the overall promotion of active aging in urban and rural areas.

(3) Launch aging products to meet the needs of the elderly

A Spanish company launched a video calling APP specifically for the elderly, eliminating the cumbersome registration process and allowing registered elderly users to log in by simply entering their names. Fifty nursing homes in Spain have applied to use the software, which has been downloaded more than 20,000 times. This kind of attention to the needs of the elderly group to explore products and services for aging transformation has important reference value for China. At present, most mobile phones in our market are designed for young people. However, there are significant obstacles in using smartphones due to the significant gap between the elderly and young people in terms of age, physical health, and cognitive ability. Especially in terms of function operation and downloading APPs, the elderly often encounter many difficulties, which bring great inconvenience to the elderly transportation, daily consumption, and other aspects and affect their physical and mental health development, reduce their sense of social participation, and inhibit the realization of active aging. The research results showed that the more Internet access devices the elderly have, the higher their social participation level and the higher their active aging level. Therefore, society should fully consider the needs of the elderly group, provide smartphones suitable for the elderly, and optimize the use of pages to make the displayed content as simple and easy to operate as possible, such as increasing readability, operation tips, voice assistance, and other functions, which are convenient for the elderly to obtain information more quickly. It can stimulate the consumption of smartphones by the elderly, enable more older people to participate in the digital age, enable the elderly group to obtain more happiness, and then realize active aging.

## Figures and Tables

**Table 1 ijerph-18-12675-t001:** The basic situation of the variable.

Variable	Mean/%	S.D.	Variable	Mean/%	S.D.
Active aging	12.90	2.16	Open the web page (%)	60.74	
Guarantee (%)	97.32		Find information (%)	57.38	
Health	19.34	3.26	Male (%)	54.03	
Social participation	12.06	2.80	Age	66.47	5.72
Life satisfaction ^1^	4.33	0.84	Education	10.88	3.28
Digital divide among young people **Independent variables**	14.14	1.15	Marriage Status (%)		
Digital divide among middle-aged people Digital divide among young people	12.9114.14	1.151.15	Married	78.19	
Base equipment Digital divide among middle-aged people	1.7212.91	1.101.15	Unmarried	3.69	
Internet usage levelBase equipment	10.241.72	1.841.10	Divorce	4.03	
Internet basic ability Internet usage level	2.8210.24	1.521.84	Widowed	14.09	
Number of Internet access devices Internet basic ability	1.722.82	1.101.52	Communist (%)	33.56	
Internet frequency (%) Number of Internet access devices	1.72	1.10	Number of properties	0.92	0.65
Frequent Internet frequency (%)	32.21		Non-Agriculture Hukou (%)	88.26	
Often Frequent	32.2132.21		Region (%)		
Sometimes Often	14.4332.21		Central	16.11	
Seldom Sometimes	12.7514.43		Eastern	76.51	
Never Seldom	8.3912.75		Westward	7.38	
Online reading time Never	95.098.39	114.50	Live alone (%)	18.79	
Electronic payment (%) Online reading time	32.8995.09	114.50	Cognitive ability	7.69	1.78
Electronic payment (%)	32.89				

^1^ Life satisfaction is measured by frustration survey indicators.

**Table 2 ijerph-18-12675-t002:** Linear regression results of indicators such as the digital divide of the elderly and active aging of the elderly.

	Health	Social Participation	Guarantee	Active Aging	Life Satisfaction
Digital divide	−0.498 ***	−0.316 *	−0.467 *	−0.335 ***	−0.164 **
	(−2.830)	(−1.952)	(−1.647)	(−2.749)	(−2.277)
Gender (Reference group: Female)					
Male	0.397	−0.455	0.706	−0.204	0.136
	(1.107)	(−1.377)	(1.495)	(−0.821)	(0.960)
Age	0.033	−0.000	−0.084 *	0.008	0.014
	(1.039)	(−0.014)	(−1.934)	(0.345)	(1.071)
Education	−0.001	0.016	0.095	0.011	0.005
	(−0.009)	(0.277)	(0.818)	(0.252)	(0.209)
Marriage (Reference group: Married)					
Unmarried	−1.031	0.192	5.133	−0.121	−0.755 **
	(−1.065)	(0.215)	(0.004)	(−0.181)	(−2.052)
Divorce	−0.852	1.438*	4.711	0.755	−0.447
	(−0.948)	(1.736)	(0.004)	(−1.215)	(−1.316)
Divorce	−1.339 **	−0.585	6.992	−0.713 *	−0.454 **
	(−2.303)	(−1.091)	(0.014)	(−1.774)	(−2.016)
Political outlook (Reference group: Masses)					
Communist	0.663 *	1.088 ***	−0.534	0.888 ***	0.054
	(1.682)	(2.997)	(−0.985)	(3.257)	(0.338)
Number of properties	0.572 **	0.190	0.705	0.269	−0.075
	(2.031)	(0.732)	(1.510)	(1.380)	(−0.655)
Account (Reference Group: Rural)					
Non-rural	0.053	1.332 **	−0.317	0.900 **	−0.066
	(0.089)	(2.441)	(−0.488)	(2.198)	(−0.289)
Region (Reference Group: Westward)					
Central	3.278 ***	−0.440	−5.625	0.503	0.561 *
	(4.211)	(−0.613)	(−0.006)	(0.935)	(1.927)
Eastern	3.441 ***	−0.723	−5.154	0.356	0.608 **
	(4.876)	(−1.112)	(−0.005)	(0.73)	(2.334)
Whether to live alone (Reference group: Living alone)					
Non-living alone	0.822	0.361	0.628	0.444	−0.533 **
	(1.625)	(0.774)	(0.905)	(1.268)	(−2.576)
Cognitive ability	−0.025	0.003	0.301 *	−0.003	0.078 *
	(−0.224)	(0.027)	(1.957)	(−0.038)	(1.761)
N	298	298	298	298	298
$R^{2}$	0.235	0.121	0.4432	0.169	0.0645

Noted: The decision coefficient of the Guarantee and Life satisfaction model was Pseudo R^2. *, **, and *** mean statistical significance at 10%, 5%, and 1%, respectively.

**Table 3 ijerph-18-12675-t003:** Linear regression results of secondary indicators of the digital ability of the elderly and active aging indicators, indicators of active aging and life satisfaction indicators of the elderly.

	Health	Social Participation	Guarantee	Active Aging	Life Satisfaction
Base equipment	0.185	0.291 *	0.684 *	0.240 **	0.210 ***
	(1.077)	(1.858)	(1.710)	(2.034)	(2.820)
$R^{2}$	0.217	0.120	0.5125	0.1593	0.0693
Internet usage level	0.225 **	0.061	0.196	0.141 **	0.064 *
	(2.251)	(0.662)	(1.555)	(2.087)	(1.650)
$R^{2}$	0.227	0.110	0.3640	0.1240	0.0579
Internet basic ability	0.405 ***	0.237 *	0.391 *	0.259 ***	0.085 *
	(3.024)	(1.920)	(1.707)	(2.789)	(1.709)
$R^{2}$	0.238	0.120	0.4486	0.1698	0.0502
Other control variables	yes	yes	yes	yes	yes
N	298	298	298	298	298

Noted: The decision coefficient of the Guarantee and Life satisfaction model was Pseudo $R^{2}$. *, **, and *** mean statistical significance at 10%, 5%, and 1%, respectively.

**Table 4 ijerph-18-12675-t004:** Linear regression results of third indicators of the digital ability of the elderly and active aging indicators, indicators of active aging and life satisfaction indicators of the elderly.

	Health	Social Participation	Guarantee	Active Aging	Life Satisfaction
Number of Internet access Devices	0.185	0.291 *	0.684 *	0.240 **	0.210 ***
	(1.077)	(1.858)	(1.710)	(2.034)	(2.820)
$R^{2}$	0.217	0.120	0.5125	0.1593	0.0693
Internet frequency	0.294 **	0.047	0.219	0.208 **	0.101 *
	(2.055)	(0.355)	(1.506)	(2.077)	(1.820)
$R^{2}$	0.225	0.109	0.2343	0.0697	0.0601
Online reading time	0.003*	0.001	0.010 *	0.001	0.001 *
	(1.673)	(0.784)	(1.781)	(1.290)	(1.733)
$R^{2}$	0.221	0.111	0.5436	0.1520	0.0509
Whether to pay electronically (Reference group: No)					
Pay electronically	0.825 **	0.739 **	0.321	0.753 ***	0.313 **
	(2.076)	(2.032)	(0.61)	(2.735)	(2.019)
$R^{2}$	0.225	0.122	0.3640	0.1472	0.0535
Whether to open the page (Reference group: No)					
Open the web page	0.786 **	0.638 *	2.080 **	0.623 **	0.248 *
	(2.054)	(1.818)	(2.142)	(2.359)	(1.707)
$R^{2}$	0.225	0.119	0.5813	0.1634	0.0385
Whether to look for information (Reference group: No)					
Find information	1.080 ***	0.140	1.311 *	0.465 *	0.260 *
	(2.800)	(0.392)	(1.893)	(1.731)	(1.774)
$R^{2}$	0.235	0.109	0.5624	0.1339	0.0358
Other control variables	yes	yes	yes	yes	yes
N	298	298	298	298	298

Noted: The decision coefficient of the Guarantee and Life satisfaction model was Pseudo $R^{2}$. *, **, and *** mean statistical significance at 10%, 5%, and 1%, respectively.

## Data Availability

Data are available on reasonable request.

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
