# Peer review of "The Digital Divide and Active Aging in China"

_ijerph, 2021, doi:10.3390/ijerph182312675_

Round 1

Reviewer 1 Report

We suggest to the author to putintroduct some data that permit to compare the percentage of elderly (and the digital divide) in China with other part of the world.

Consequently, in the last part, it could be interesting to discuss the solutions proposed with measures realised by other countries.

Author Response

We are grateful to the editor for the opportunity to revise the paper to reach the publishable standard. The reviewer's comments appear in italics, while our responses are in normal font. Our adjusted discussions are with yellow marks.

Point 1:We suggest to the author to putintroduct some data that permit to compare the percentage of elderly (and the digital divide) in China with other part of the world.

Response 1:Thank you very much for making this suggestion. As you said, it is necessary for us to compare the percentage of elderly and the digital divide between China and the rest of the world, because it can help us better understand the situation in China.Therefore, in the literature review section, we add some data to compare the proportion of the elderly and the gap of the digital divide of the elderly between China and other part of the world. For details, please see lines 129-142 of the attachment paper in PDF format.

Point 2: Consequently, in the last part, it could be interesting to discuss the solutions proposed with measures realised by other countries.

Response 2: Thank you for your suggestion. As you said, it is interesting and helpful to discuss the proposed solutions in terms of measures that have been implemented in other countries. Data show that compared with other regions in the world, China's aging level is relatively low, but the elderly population is large, and the digital divide between the elderly is prominent. Therefore, the measures taken by other countries and regions to deal with the digital divide of the elderly have certain reference value and significance for China. Therefore, we refer to solutions of other countries on the basis of empirical results when making suggestions at last. For details, please see lines 472-481, 508-520 and 539-544 of the attachment paper in PDF format.

Reviewer 2 Report

This is a cross-sectional study that aims at measuring the digital divide in a sample of Chinese seniors, not only in qualitative but also in quantitative terms.

Overall, the work is interesting because it provides information on a very dynamic society like the Chinese one, although it does not lead to original conclusions except for the attempt to give us quantitative measures.

My observations:

  1. It is not clear to me how the physical and mental health indicator was calculated: it refers to unspecified physical health parameters (use of drugs? Use of health services? Comorbidities?) and for the psychological aspects to dimensions that are very difficult to measure (happiness and sense of loneliness); finally, these two aspects are combined together according to unclear criteria; furthermore, no mention is made of the participants' cognitive functions, which are of primary interest in a geriatric setting.
  2. It is not even clear to me how the relative weight of physical and mental health, social participation and safety was calculated in composing the overall indicator of "active aging".
  3. Given the nature of the study, I disagree with the sentence in the conclusions section “The digital divide of the elderly has a significant inhibitory effect on their active aging level”. It seems more appropriate to say that the digital divide is associated with a lower level of active aging.
  4. Likewise and for the same reasons, the next two statements cannot be accepted as true: “By decomposing the digital divide, it is found that various indicators of the digital technology ability of the elderly have effectively promoted active aging” and that digital divide “inhibit(s) the realization of their active aging”.

Author Response

We are grateful to the editor for the opportunity to revise the paper to reach the publishable standard. The reviewer's comments appear in italics, while our responses are in normal font. Our adjusted discussions are with yellow marks.

Point 1(a): It is not clear to me how the physical and mental health indicator was calculated: it refers to unspecified physical health parameters (use of drugs? Use of health services? Comorbidities?) and for the psychological aspects to dimensions that are very difficult to measure (happiness and sense of loneliness); finally, these two aspects are combined together according to unclear criteria.

Response 1(a): Thank you very much for making this suggestion. We selected the physical and mental health dimensions from both physical and psychological aspects, and finally selected five indicators, happiness and loneliness indicators directly from the CGSS data center to choose, Learn from Jin Yongai, this paper will add up the value of 5 indicators, the higher the score indicates the better physical and mental health. For detailed revisions, please see lines 210-221 of the attachment paper in PDF format. 

Point 1(b): No mention is made of the participants' cognitive functions, which are of primary interest in a geriatric setting.

Response 1(b): Thanks for your suggestion. Based on CGSS data, we selected older adults' cognitive ability indicators, put them in the model as control variables, and found that older people with higher cognitive ability scores experienced fewer difficulties in their lives, as shown in lines 309-311 and 373-375 of the attachment paper in PDF format. 

Point 2: It is not even clear to me how the relative weight of physical and mental health, social participation and safety was calculated in composing the overall indicator of "active aging".

Response 2: Thanks for your suggestion. After constructing a comprehensive evaluation index system for active aging, the next step is to empower each index. The size of the indicator's weight directly determines the importance of the indicator and affects the final result. Therefore, how to choose a scien-tific and effective method of empowerment is very important. At present, the commonly used methods of empowerment mainly include personal empowerment and objective empowerment.

The subjective empowerment method is based on expert opinion, which mainly depends on the expert's experience and knowledge reserved for many years. However, subjective empowerment law relies too much on people's subjective judgment, and the quality of experts is uneven, easy to affect the evaluation results. Subjective empowerment law mainly includes hierarchical analysis and expert opinion.

The objective empowerment method is based on the intrinsic relationship of objective values to calculate the index weight of the empowerment method, not affected by subjective judgment, and the result is more objective than others. Commonly used objective empowerment methods include factor analysis, principal component analysis, neural net-work, entropy.

Compared with other empowerment methods, the entropy power method can make full use of systematic information, consider the ambiguity and uncertainty of evaluation criteria and indicators, and effectively reduce the subjectivity in evaluation. Therefore, this paper uses the entropy right method as the empowerment method of the positive aging comprehensive evaluation, calculates the final score, and makes the evaluation result more accurate.

Active aging comprises three indicators: physical and mental health, social participation, and security, synthesized by the entropy weight method, and the higher the value, the higher the degree of active aging of the elderly population. 

Point 3: Given the nature of the study, I disagree with the sentence in the conclusions section “The digital divide of the elderly has a significant inhibitory effect on their active aging level”. It seems more appropriate to say that the digital divide is associated with a lower level of active aging. 

Response 3: Thanks for your suggestion. We have revised the wording of the sentence to read, "There is a significant negative correlation between the digital divide among older persons and China's active aging." For detailed revisions, please see lines 422-423 of the attachment paper in PDF format. Thank you.

Point 4: Likewise and for the same reasons, the next two statements cannot be accepted as true: “By decomposing the digital divide, it is found that various indicators of the digital technology ability of the elderly have effectively promoted active aging” and that digital divide “inhibit(s) the realization of their active aging”. 

Response 4: Thank you very much for your suggestion. We changed the tone of the sentence to “By breaking down the digital divide, it is found that there is a positive correlation between indicators of the digital technological capacity of older persons and active aging ” and “It brings adverse effects on the physical and mental health, social participation, and security of the elderly, thus not conducive to active aging.”, respectively. For detailed revisions, please see lines 427-429 and 456-460 of attachment the paper in PDF format. Thank you.
